# Partially Aligned Cross-modal Retrieval via Optimal Transport-based Prototype Alignment Learning

## ABSTRACT

Supervised cross-modal retrieval (CMR) achieves excellent performance thanks to the semantic information provided by its labels, which helps to establish semantic correlations between samples from different modalities. However, in real-world scenarios, there often exists a large amount of unlabeled and unpaired multimodal training data, rendering existing methods unfeasible. To address this issue, we propose a novel **partially aligned cross-modal retrieval method called Optimal Transport-based Prototype Alignment Learning (OTPAL).** Due to the high computational complexity involved in directly establishing matching correlations between unannotated unaligned cross-modal samples, instead, we establish matching correlations between shared prototypes and samples. To be specific, we employ the optimal transport algorithm to establish cross-modal alignment information between samples and prototypes, and then minimize the distance between samples and their corresponding prototypes through a specially designed prototype alignment loss. As an extension of this paper, we also extensively investigate the influence of incomplete multimodal data on cross-modal retrieval performance under the partially aligned setting proposed above. To further address the above more challenging scenario, we raise a scalable prototype-based neighbor feature completion method, which better captures the correlations between incomplete samples and neighbor samples through a cross-modal self-attention mechanism. Experimental results on four benchmark datasets show that our method can obtain satisfactory accuracy and scalability in various real-world scenarios.

## CCS CONCEPTS

• **Information systems → Multimedia and multimodal retrieval**.

## KEYWORDS

Robust Cross-modal Retrieval, Partially Aligned Data, Optimal Transport Strategy, Prototype Alignment Learning

## 1 INTRODUCTION

Cross-modal retrieval (CMR) is a cross-modal search task: retrieving relevant information from different modalities, such as text, image, audio, etc. With the introduction of deep learning, cross-modal retrieval has made great progress. It has become the core research in many multi-modal applications, such as audio-video retrieval [30], automatic story generation [38], visual question answering [16, 23, 41], and medical image-report retrieval [3, 37]. Thus, CMR has attracted increasing attention in both academia and industry due to its importance in real-world applications.

Common cross-modal retrieval methods include the following types: (1) unsupervised cross-modal retrieval (US-CMR) [7, 8, 35, 36, 39], (2) supervised cross-modal retrieval (S-CMR) [20, 29, 33, 34, 42, 43, 47] and (3) semi-supervised cross-modal retrieval (SS-CMR)

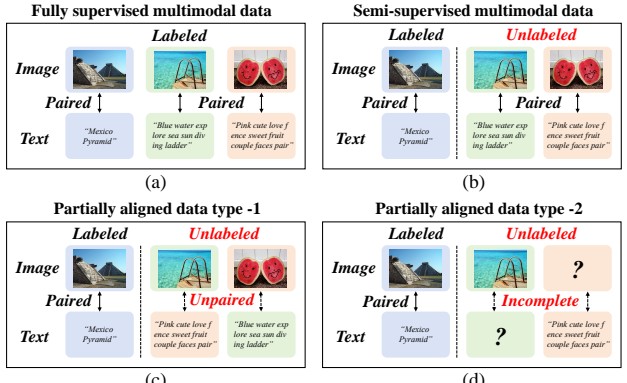

Figure 1: The differences between our proposed partially aligned CMR and traditional methods. (a) In traditional supervised CMR, images and texts are labeled and paired. (b) The semi-supervised CMR method operates on both partially labeled paired and partially unlabeled paired data. To enhance the practicality of the aforementioned CMR methods in open environments, we introduce two types of partially aligned CMR methods tailored for unlabeled unpaired data (c) and unlabeled incomplete data (d), respectively.

[12, 17, 18]. S-CMR methods can access semantic labels, which are generally superior to US-CMR performance. However, it usually incurs substantial labeling costs. Thus, SS-CMR methods [12, 17, 18] have been proposed to balance the conflict between the performance and cost of labeling data.

Semi-supervised methods [12, 17, 18] exploit the intrinsic structure, pairwise relationships, and semantic information between different modalities to improve retrieval performance on the overall data. However, despite existing semi-supervised cross-modal methods performing well, most of them require pairwise data to establish representation consistency [17] and prediction consistency [18], which is only applicable to well-paired data. For instance, as depicted in Fig. 1 (b), even though some of the images and texts in the aforementioned semi-supervised methods are unlabeled, they still leverage the pairwise information of the images and texts. Unfortunately, in real-world applications, this assumption is always violated due to the diversity of multimodal data sources and noise issues, the pairwise correspondences are also partially available and inevitably suffer from partial mismatch problems. For example, images collected from web pages are difficult to be semantically related to text descriptions, or they can easily be mistakenly collected as irrelevant image-text pairs (as shown in Fig. 1 (c)). As a result, real-world multi-modal data often consists of a small piece of well-paired multi-modal data and a massive amount of unpaired multi-modal data. Furthermore, in addition to irrelevant image-text pairs resulting from erroneous acquisition, multimodal pairwise

data are inevitably limited by complete requirements. Yet this requirement is also equally demanding. For instance, in practical scenarios, multimodal training datasets inevitably suffer damage or loss during data collection, processing, storage, and transmission. Therefore, incomplete modal data will result in some pairwise relationships being unavailable (as shown in Fig. 1 (d)). We unify the above two problems, referring to them collectively as cross-modal retrieval under two types of partially aligned data.

To tackle the above challenges of two types of partially aligned data (Fig. 1 (c) and (d)) in cross-modal retrieval, we propose an **Optimal Transport-based Prototype Alignment Learning (OTPAL)** framework, which comprises a dual optimal transport-based prototype alignment module, by leveraging semantic prototypes learned from labeled data to facilitate the alignment of unlabeled unpaired data with prototypes. Specifically, we initially employ multimodal classification and invariance learning to capture modality-discriminative and invariant representations on labeled paired data. Furthermore, since the prototype can be characterized as the semantic center of each category in the feature space and can serve as the bridge between data and semantics, we define a prototype for each category and learn and update it through training. Besides, to learn the alignment relationship between unpaired data and prototypes, minimizing intra-class variation, and maximizing inter-class distance, our OTPAL tackles the partially aligned problem caused by unlabeled unpaired data (Fig. 1 (c)) from the perspective of an optimal transport strategy, leveraging the inherent correlation among multi-modal data to construct the effective transport cost, and progressively transport unlabeled unpaired samples into the correct prototype at the minimum cost. Finally, to address the partial alignment problem caused by unlabeled incomplete data (Fig. 1 (d)) and further extend the application of partially aligned cross-modal retrieval, we construct a prototype-guided neighbor-based feature completion method, which can perform feature reconstruction based on the correlation between missing features and neighbor features. **The major contributions of this paper can be summarized as follows:**

- We clarify two key challenges in enhancing the robustness of cross-modal retrieval systems in practical applications. Furthermore, we propose a prototype-based optimal transport learning strategy for partially aligned cross-modal retrieval.
- We propose a novel framework for partially aligned cross-modal retrieval, which minimizes the distance between samples and their corresponding prototypes, thereby enhancing modality discriminability and invariance.
- We design a scalable cross-modal prototype and neighbor completion method to address incomplete partially aligned intractable scenarios.
- Experimental results on four benchmark datasets demonstrate the effectiveness of our method, which can handle cross-modal retrieval tasks in various complex scenarios.

## 2 RELATED WORK

Cross-modal retrieval (CMR) methods can be roughly divided into two groups: traditional multimodal representation learning methods and deep multimodal representation learning methods.

**Traditional CMR methods** primarily learn linear or simple nonlinear mappings through statistical analysis. Canonical Correlation Analysis (CCA) [8] is one of the most typical unsupervised subspace learning methods, which leverages canonical correlation analysis to maximize pairwise correlations between two sets of heterogeneity. Many CCA-based extensions and similar methods have been proposed, such as Kernel Canonical Correlation Analysis (KCCA) [36] and Partial Least Squares (PLS) [28]. To learn the common space with semantic information, a joint representation learning (JRL) [44] is proposed to jointly explore the correlation and semantic information in a unified optimization framework. To exploit the correlations on semi-supervised data, Generalized Semi-supervised Structured Subspace Learning (GSS-SL) [45] is proposed to take the label space as a linkage to model the correlations among different modalities.

**Deep learning CMR methods** exploit the power of deep neural networks to capture non-linear relationships. DAVAE [14] proposes dual-aligned variational autoencoders to learn latent common representations in the view of distribution level and semantic level. To learn intra-modality and inter-modality representation correlations, several cross-modal retrieval methods [19, 21, 22] have been proposed. More recent works [29, 34, 42, 43, 47] exploit pair-wise label and class-wise information to learn modality invariance and discriminability. PAN [43] learns a unified prototype for each semantic category and uses them as anchors to learn cross-modal representations. Cross-Modality Cross-Instance Contrastive Learning for Cross-Media Retrieval ($C^3CMR$) [34] proposes intra-modal and inter-modal contrastive learning to capture data associations and enhance the discriminative capability of features. Incomplete Cross-Modal Retrieval with Deep Correlation Transfer (ICMR-DCT) [29] proposes to model incomplete multi-modal data and dynamically capture adjacency semantic correlation for cross-modal retrieval.

To balance the conflict between the performance and cost of labeling data, some semi-supervised works [12, 17, 18] are proposed. Label Prediction Framework (LPF) [18] predicts the labels of the unlabeled data utilizing complementary information from both modalities for semi-supervised cross-modal retrieval. Semi-Supervised Multi-Modal Learning with Balanced Spectral Decomposition (SMLN) [12] correlates different modalities by capturing the intrinsic structure and discriminative correlations of multimedia data. However, the role of semantic information is not considered. Self-Supervised Correlation Learning (SCL) [17] leverages label prediction and class-aware contrastive learning to learn modality invariance and discrimination. Although the above works perform well on semi-supervised data, it is difficult to extend to partially aligned data.

**Optimal Transport Strategy** (OT) was originally proposed to depict the distance between two probability distributions, which is often used to find correspondences with learnable features or measure distribution distances. It has been applied to many fields, such as domain adaptation [5], sequence alignment [46], vision and language [2]. Cuturi [6] first proposed to use the Sinkorn algorithm to compute an approximate transport coupling with an entropic regularization. This method is lightspeed and can handle large-scale problems efficiently. Unbalanced Optimal Transport [1] (UOT) is proposed to relax equality constraint and the corresponding optimization problem can be reformulated as a non-negative

penalized linear regression problem. In contrast, we apply OT for a data-to-prototype assignment problem.

## 3 PROPOSED METHOD

The purpose of this paper is to train a model that can effectively address the two types of partially aligned cross-modal retrieval we introduce. To this end, we propose that the Optimal Transport-based Prototype Alignment Learning (OTPAL) model (Fig. 3) consists of three modules: 1) Multimodal Classification and Invariance Learning (MCIA) in Section 3.2 can help the model capture basic discriminant representation and modality invariance. 2) Dual Optimal Transport-based Prototype Alignment (DOTPA) in Section 3.3 can effectively establish the correspondence between unlabeled unpaired images and texts. 3) Prototype-based Neighborhood Feature Completion (PNFC) in Section 3.4 aims to generate incomplete features to maintain the semantic consistency of multimodal data.

### 3.1 Problem Formulation

**Notations.** Different from traditional cross-modal retrieval, we propose a more practical task: partially aligned cross-modal retrieval, aimed at addressing the challenges of partially unaligned and partially incomplete multimodal data commonly encountered in real-world scenarios. Next, we formulate the partially aligned problem concerning the image and text modalities. We assume that two types of partially aligned data are illustrated in Fig. 2. **Data type-1** consists of two parts: 1) labeled aligned image and text pairs $\mathcal{D}^l$, and 2) unlabeled unaligned data $\mathcal{D}^u_u$. **Data type-2** contains two parts of data: 1) labeled aligned images and texts $\mathcal{D}^l$, and 2) unlabeled incomplete multimodal data $\mathcal{D}^u_m$. Here, $\mathcal{D}^l = \{x^v_i, x^t_i, y_i\}^{N_l}_{i=1}$, where images and texts are paired via semantic labels $y_i$ and $N_l$ is the number of training samples from the labeled data. $\mathcal{D}^u_u = \{x^v_i, x^t_i\}^{N_u}_{i=1}$, where the image $x^v_i$ and text $x^t_i$ are not aligned. $\mathcal{D}^u_m = \{\mathcal{D}^v, \mathcal{D}^t\}$, where $\mathcal{D}^v = \{x^v_i\}^{N_v}_{i=1}$ defines only image modal data, while the corresponding text is inaccessible, and $\mathcal{D}^t = \{x^t_i\}^{N_t}_{i=1}$ denotes only text modal data, while the corresponding image is inaccessible. $N_v$ and $N_t$ represent the number of image-only samples and text-only samples respectively. The goal of OTPAL is to establish a common space for partially aligned data. In this space, the representations of images and texts can be better semantically aligned.

### 3.2 Multimodal Classification and Invariance Learning

We first explore a type of partially aligned cross-modal retrieval method that we propose, shown in Fig. 2 (a). In the labeled data in Fig. 2 (a), images and texts are paired with semantic labels one-to-one. However, in the unlabeled data, image samples and text samples are not aligned (unpaired) and lack semantic information. Next, we specify our proposed method.

Due to the heterogeneous properties of different modalities, cross-modal retrieval first learns a common representation space. Thus, the dual-stream backbones (e.g., CLIP-RN50 [24] and CLIP-Xformer [24]) $f_v(.)$ and $f_t(.)$ are used to extract the image $x^v_i$ and text $x^t_i$ modal features, and then project the features from different modalities into a modality-shared feature space formulated as:

$$z^v_i = f_v(CLIP_v(x^v_i); \theta_v), \tag{1}$$

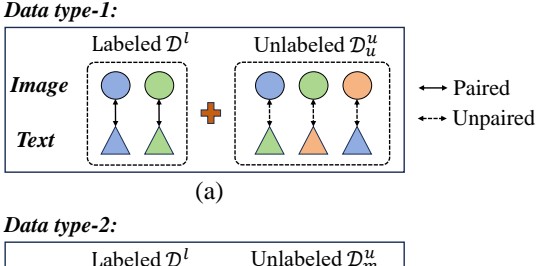

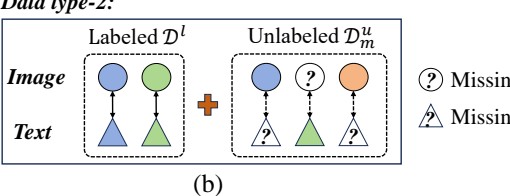

**Figure 2: Illustration of partially aligned multimodal data.**

$$z^t_i = f_t(CLIP_t(x^t_i); \theta_t), \tag{2}$$

where $CLIP_v(.)$ and $CLIP_t(.)$ define visual and textual encoders. To generate a common representation space, two modality-specific multi-layer perception (MLP) $f_v(.)$ and $f_t(.)$ are utilized, where $\theta_v$ and $\theta_t$ are trainable parameters of modality-specific multi-layer perception. Finally, we obtain the representation features $z^v_i$ and $z^t_i$ projected on a common subspace. Multimodal Classification and Invariance Learning (MCIA) over the labeled data can help the model learn basic discriminative representation ability. At first, we design two modality-specific classifiers based on the projected features, and then utilize cross-entropy loss and standard triplet loss [33] to optimize the network parameters expressed as follows:

$$L^l_{ce} = -\frac{1}{N_l} \sum^{N_l}_{i=1} (y_i \cdot \log(\hat{y}^v_i) + y_i \cdot \log(\hat{y}^t_i)), \tag{3}$$

$$L^l_{tri} = \frac{1}{|X_v|} \sum_{(z^v_i, z^{t+}_j, z^{t-}_j) \in X_v} [d(z^v_i, z^{t+}_j) - d(z^v_i, z^{t-}_j) + \delta] \tag{4}$$

$$+ \frac{1}{|X_t|} \sum_{(z^t_i, z^{v+}_j, z^{v-}_j) \in X_t} [d(z^t_i, z^{v+}_j) - d(z^t_i, z^{v-}_j) + \delta], \tag{5}$$

where $\hat{y}^v_i$ and $\hat{y}^t_i$ are predicted values by the two MLP classifier layers. $X_v$ denotes the set of triplets by selecting $z^v_i$ as the anchor to find the positive text $z^{t+}_j$ and the negative text $z^{t-}_j$. The same applies to $X_t$. $|X_v|$ and $|X_t|$ are their cardinalities.

The unsupervised classification loss $L^u_{ce}$ is defined as the cross-entropy between the pseudo-labels $\check{y}_i$ and the model's predictions $\hat{y}^v_i, \hat{y}^t_i$ in the image and text modalities, respectively:

$$L^u_{ce} = -\frac{1}{N_u} \sum^{N_u}_{i=1} (\check{y}_i \cdot \log(\hat{y}^v_i) + \check{y}_i \cdot \log(\hat{y}^t_i)). \tag{6}$$

The MCIA loss function (Fig. 3) combines $L^l_{ce}$, $L^l_{tri}$ and $L^u_{ce}$, which helps the model to learn certain intra-modality discriminability and inter-modal alignment representations and can be represented as:

$$L_{MCIA} = L^l_{ce} + L^l_{tri} + \beta L^u_{ce}. \tag{7}$$

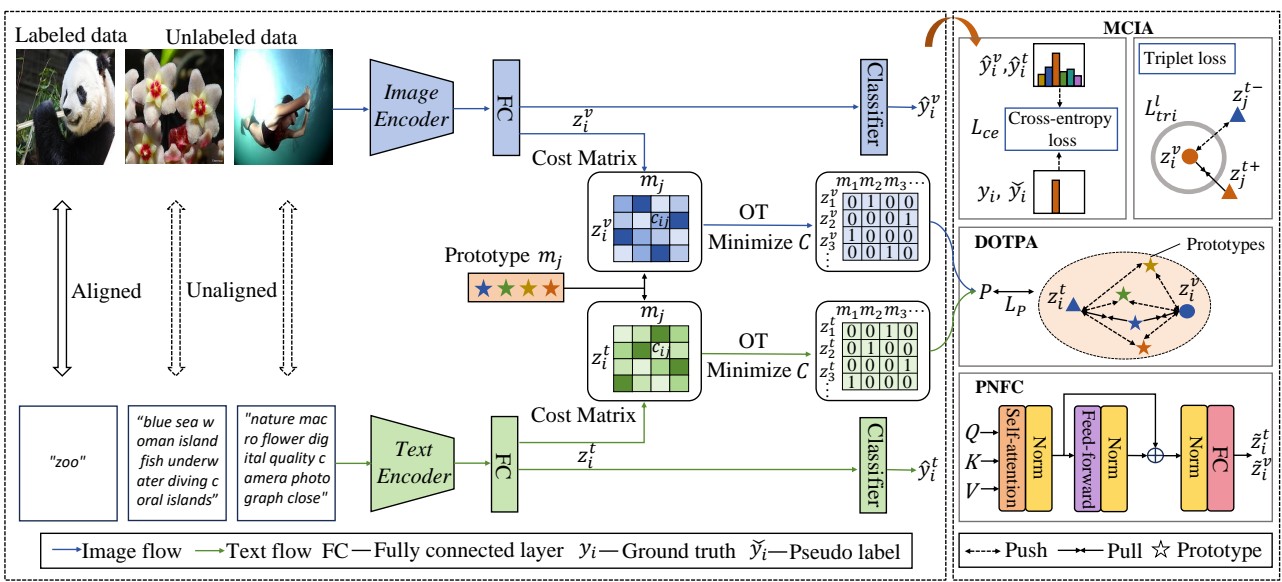

**Figure 3: Overview of the OTPAL framework. To mitigate modal heterogeneity and establish cross-modal semantic correspondence in partially aligned multimodal scenarios, the OTPAL model is introduced, comprising three main components: Multimodal Classification and Invariance Learning (MCIA) in Section 3.2, Dual Optimal Transport-based Prototype Alignment (DOTPA) in Section 3.3, and Prototype-based Neighborhood Feature Completion (PNFC) in Section 3.4.**

## 3.3 Dual Optimal Transport-based Prototype Alignment

The aforementioned MCIA does not directly explore the alignment relationship of unlabeled unpaired data from two modalities, thus it cannot address situations where modality data is partially aligned. Next, we focus on the key challenge of the partially aligned CMR task: unlabeled unpaired image and text data association. To address such an issue, we propose a Dual Optimal Transport-based Prototype Alignment (DOTPA) method (Fig. 3). It simultaneously matches the unpaired images and texts with the prototypes and obtains a matching matrix. The prototype [40] can be regarded as a set of modality-shared semantic centers, and features with similar semantics have consistent prototype assignments. The shared prototype for images and texts can be expressed as:

$$M = \{m_j \mid j = 1, ..., K\}, \tag{8}$$

where $j$ denotes the index of the category. During the initialization phase, we randomly generate a set of prototypes that are learned and updated through training. We use prototypes as an intermediate bridge to find correspondences between unlabeled unpaired images and texts.

Specifically, for the given $N_u$ suppliers (unlabeled unpaired images and texts) and $K$ demanders (prototypes). The supplier supplies unlabeled unpaired images and texts to the demander, described as a vector $\mathbf{a}$, and the demander receives unlabeled unpaired images and texts from the supplier, described as a vector $\mathbf{b}$. We formulate the cross-modal prototype assignment task as an optimal transport problem. The problem is to find an optimal transportation plan $P \in R^{B \times K}$ to minimize the transport cost $C$, which satisfies the following equation:

$$P^* = \arg\max_{P \in \mathcal{P}} \langle P, C \rangle_F + \lambda H(P), \tag{9}$$

where $\langle P, C \rangle_F$ is the Frobenius inner product between the cost matrix $C$ and the matching plan $P$. $C$ is a cost matrix where each element $C_{ij}$ represents the similarity from $z_i^v$ ($z_i^t$) to $m_j$, and the cost matrix for our optimal transport problem can be denoted $C_{ij} = 1 - \cos(z_i^v, m_j)$. We add entropy regularization on $P$ as $H(P) = \sum_{ij} P_{ij} \log P_{ij}$. This ensures that $P$ is not over-concentrated on a few elements. Finally, the solution of Eq. (9) can constrain a transportation polytope:

$$\mathcal{P} = \{P \in R^{B \times K} | P\mathbf{1}_K = \mathbf{a}, P^T\mathbf{1}_B = \mathbf{b}\}, \tag{10}$$

where $P_{ij}$ means the transport plan between the $i - th$ unlabeled unpaired sample and the $j - th$ prototype, and $P$ contains all non-negative $B \times K$ elements, with row and column sums equal to $\mathbf{a}$ and $\mathbf{b}$, respectively. We preserve the assignment matrix $P^*$ and the solution of the transportation polytope, solved efficiently using the Sinkhorn-Knopp algorithm [6], which can be written as follows:

$$P^* = Diag(u) \exp(\frac{1}{\lambda}C) Diag(v), \tag{11}$$

where $u, v$ are row and column normalized vectors and can be calculated through the iterative Sinkhorn-Knopp algorithm [6].

To minimize intra-class and maximize inter-class variations, and learn discriminative representations, each sample should be close to the prototype to which it belongs. Specially, for each feature $z_i^r, r = \{v, t\}$, we can calculate the softmax of its similarity to the corresponding prototype, formulated as follows:

$$S_{ij} = \frac{\exp(sim(z_i^r, m_j)/T)}{\sum_{j=1}^K \exp(sim(z_i^r, m_j)/T)}, \tag{12}$$

where $sim(.)$ represents the cosine similarity, $T$ is the temperature factor, and $K$ is the number of shared prototypes. We then define

the data-to-proxy alignment loss via the cross-entropy loss as:

$$L_{pr}^l = -\frac{1}{N_l} \sum_{i=1}^{N_l} \sum_{j=1}^{K} \mathbf{1}(y_i = j) \log(S_{ij}), \quad (13)$$

where $K$ is the number of classes and $N_l$ denotes the number of labeled data. Analogously, to effectively leverage a large amount of unlabeled data, we generalize Eq. (13) to unlabeled unpaired image and text data, establishing their matching associations with shared semantic prototypes, which can be calculated as follows:

$$L_{pr}^u = -\frac{1}{N_u} \sum_{i=1}^{N_u} \sum_{j=1}^{K} \mathbf{1}(P_{ij} = j) \log(S_{ij}), \quad (14)$$

where $P_{ij}$ is the optimal transport plan obtained by Eq. (9). To obtain a more reliable prototype assignment, we use a prototype-based reliability measurement method. The reliability of the $P_{ij}$ is measured according to the similarity between the sample and its class prototype, and we select the threshold of unpaired image and text similarity $R(P)$ to be greater than $\tau$ to train the DOTPA module, where $R(P_{ij}) = d(z_i^r, m_j), r = \{v, t\}$. The DOTPA method is well-suited for our proposed partially aligned cross-modal retrieval for the following reasons: (1) We align unlabeled unpaired images and texts with the prototypes to which they belong. Compared with directly aligning unlabeled unpaired images with texts, it can effectively reduce storage space and calculation amount. (2) Prototype-based alignment loss can further reduce intra-class variation and increase inter-class distance, which is conducive to learning semantic discriminative representations on both labeled and unlabeled data.

## 3.4 Prototype-based Neighborhood Feature Completion

As an extension of this paper and equally important, we delve into another partially aligned cross-modal retrieval method, as illustrated in Fig. 2 (b). The unlabeled incomplete data consists of two types: solely image data where the corresponding text is inaccessible, and solely text samples where the corresponding image samples are inaccessible. To address the aforementioned issue of unlabeled incomplete data, we propose the Prototype-based Neighborhood Feature Completion (PNFC) method (Fig. 3) to reconstruct the missing modality features and further improve the robustness of partially aligned cross-modal retrieval. The idea behind PNFC is that instances from similar semantics are expected to be consistent in modality-shared space. By incorporating neighborhood feature information with semantic prototypes, both modality complementary information and prototype semantic information can be preserved in the recovered data. To reconstruct the missing text features $\widetilde{z}_i^t$ for available image features $z_i^v$ and preserve semantic consistency between incomplete modality data, we first design $N_k(z_i^v)$ as a list of the $k$ nearest neighbors in the corresponding image modality, by using $z_i^v$ as the query to rank their distances as:

$$N_k(z_i^v) = \left[z_1^v, ..., z_k^v\right], \quad (15)$$

where $k$ is a hyper-parameter. We transfer the nearest neighbors' relation to text modality and can infer missing text feature neighbors $N_k(\widetilde{z}_i^t) = \left[z_1^t, ..., z_k^t\right]$ from $N_k(z_i^v)$. Since the prototypes are trained

with the discrimination loss, they can provide shared semantic information across modalities. Thus, we leverage the prototype and nearest neighbor features to complete the missing text features:

$$\widetilde{z}_i^t = \varphi(m_j, N_k(z_i^v), N_k(\widetilde{z}_i^t); \theta_w), \quad (16)$$

where $\theta_W$ is the trainable parameters. $\varphi$ denotes the cross-modal self-attention encoder that attends to the correlations between the complete modality feature and neighbor samples. The self-attention encoder can be expressed as:

$$\hat{z}_i^t = Attention(Q, K, V), \quad (17)$$

$$\hat{z}_i^t = FFN(LN(\hat{z}_i^t)) + LN(\hat{z}_i^t), \quad (18)$$

$$\hat{z}_i^t = LN(\hat{z}_i^t), \quad (19)$$

where $Q = z_i^v W_Q$, $K = Z^v W_K$ and $V = Z^t W_V$, $z_i^v$ is corresponding image feature of the missing modality feature $\widetilde{z}_i^t$. $Z^v$ and $Z^t$ are sets of $N_k(z_i^v)$ and $N_k(\widetilde{z}_i^t)$ neighbor sample features, and $N_k(\widetilde{z}_i^t)$ of $z_i^t$ can be inferred from $N_k(z_i^v)$. $LN()$ is the layer normalization. $FFN()$ is a feed-forward network [32]. Finally, we perform a fully-connected decoder to generate missing feature as follows:

$$\widetilde{z}_i^t = D(\hat{z}_i^t; \theta_D), \quad (20)$$

where $D(.; \theta_D)$ denotes the fully-connected decoder. To maintain representation consistency between the completion features and the corresponding available features, we propose the missing modal feature completion loss computed as:

$$L_{MFC} = \|\widetilde{z}_i^t - z_i^v\|_2^2 + \|\widetilde{z}_i^v - z_i^t\|_2^2. \quad (21)$$

**Overall Loss Function.** The overall objective $L$ is combined with three losses, including the discrimination loss $L_{MCIA}$, prototype alignment loss $L_P$ and missing feature completion loss $L_{MFC}$:

$$L = L_{MCIA} + \alpha L_P + L_{MFC}, \quad (22)$$

where $\alpha$ is the hyper-parameter to balance the different losses. The prototype alignment loss $L_P$ is the sum of $L_{pv}^l$, $L_{pt}^l$, $L_{pv}^u$ and $L_{pt}^u$.

## 4 EXPERIMENTS

### 4.1 Datasets and New Data Splitting

We undertake comprehensive experiments on four widely used cross-modal retrieval benchmarks including NUS-WIDE-10K [4], Wikipedia [26], XmediaNet [22], and Pascal-Sentence [25].

**Data type-1:** In our proposed partially aligned cross-modal retrieval, we randomly select 20%, 40% and 60% instances from the multi-modal training set as labeled aligned data, while the rest are designated as unlabeled unaligned data (Fig. 2 (a)). In the labeled aligned data, images and texts are aligned and annotated with class information. Meanwhile, in unlabeled unaligned data, images and texts are not aligned. Note that having only 20% labeled aligned and 80% unlabeled unaligned training data poses the greatest challenge in partially aligned cross-modal retrieval.

**Data type-2:** As shown in Fig. 2 (b), in unlabeled unaligned data, the image sample is available but the corresponding text is missing, or the text is accessible but the corresponding image is missing. For example, we denote 10% as labeled aligned data and 90% as unlabeled unaligned data, where 90% of unlabeled unaligned data comprises 45% image-only samples and 45% text-only samples represented as (10%L, 45%I, 45%T). Similarly, we also define various other ratios. Note that (10%L, 45%I, 45%T) is also the most challenging.

Table 1: Retrieval performance for mAP scores compared to existing methods on the NUS-WIDE-10K dataset with data type-1.

| Type | Methods | Venue | 20% Label data | | | 40% Label data | | | 60% Label data | | |
|---|---|---|---|---|---|---|---|---|---|---|---|
| | | | I2T | T2I | Avg | I2T | T2I | Avg | I2T | T2I | Avg |
| US-CMR | DCCA [39] | arXiv15 | 0.462 | 0.471 | 0.466 | 0.469 | 0.483 | 0.476 | 0.470 | 0.484 | 0.477 |
| | DCCAE [35] | PMLR15 | 0.461 | 0.473 | 0.467 | 0.469 | 0.488 | 0.478 | 0.471 | 0.487 | 0.479 |
| S-CMR | DSCMR [47] | CVPR19 | 0.591 | 0.594 | 0.592 | 0.609 | 0.603 | 0.606 | 0.618 | 0.614 | 0.616 |
| | DAVAE [14] | MM20 | 0.541 | 0.517 | 0.529 | 0.518 | 0.550 | 0.534 | 0.552 | 0.566 | 0.559 |
| | PAN [43] | SIGIR21 | 0.580 | 0.590 | 0.585 | 0.590 | 0.603 | 0.596 | 0.596 | 0.608 | 0.602 |
| | C$^3$CMR [34] | MM22 | 0.574 | 0.580 | 0.577 | 0.584 | 0.597 | 0.590 | 0.596 | 0.604 | 0.600 |
| | ICMR-DCT [29] | TOMM23 | 0.551 | 0.511 | 0.531 | 0.559 | 0.553 | 0.556 | 0.572 | 0.571 | 0.571 |
| SS-CMR | SMLN [12] | AAAI20 | 0.578 | 0.598 | 0.588 | 0.591 | 0.609 | 0.600 | 0.600 | 0.627 | 0.613 |
| | SCL$_{ss}$ [17] | TMM22 | 0.607 | 0.613 | 0.610 | 0.611 | 0.619 | 0.615 | 0.612 | 0.622 | 0.617 |
| | OTPAL | - | **0.631** | **0.639** | **0.635** | **0.636** | **0.638** | **0.637** | **0.635** | **0.645** | **0.640** |

Table 2: Retrieval performance for mAP scores compared to existing methods on the Wikipedia dataset with data type-1.

| Type | Methods | Venue | 20% Label data | | | 40% Label data | | | 60% Label data | | |
|---|---|---|---|---|---|---|---|---|---|---|---|
| | | | I2T | T2I | Avg | I2T | T2I | Avg | I2T | T2I | Avg |
| US-CMR | DCCA [39] | arXiv15 | 0.444 | 0.437 | 0.440 | 0.453 | 0.443 | 0.448 | 0.466 | 0.447 | 0.456 |
| | DCCAE [35] | PMLR15 | 0.433 | 0.418 | 0.425 | 0.445 | 0.437 | 0.441 | 0.449 | 0.441 | 0.445 |
| S-CMR | DSCMR [47] | CVPR19 | 0.541 | 0.528 | 0.534 | 0.576 | 0.563 | 0.569 | 0.590 | 0.573 | 0.581 |
| | DAVAE [14] | MM20 | 0.431 | 0.383 | 0.407 | 0.533 | 0.507 | 0.520 | 0.547 | 0.511 | 0.529 |
| | PAN [43] | SIGIR21 | 0.541 | 0.525 | 0.533 | 0.562 | 0.548 | 0.555 | 0.577 | 0.561 | 0.569 |
| | C$^3$CMR [34] | MM22 | 0.499 | 0.481 | 0.490 | 0.530 | 0.518 | 0.524 | 0.553 | 0.535 | 0.544 |
| | ICMR-DCT [29] | TOMM23 | 0.452 | 0.399 | 0.425 | 0.548 | 0.520 | 0.534 | 0.560 | 0.539 | 0.549 |
| SS-CMR | SMLN [12] | AAAI20 | 0.545 | 0.528 | 0.536 | 0.605 | 0.581 | 0.593 | 0.608 | 0.579 | 0.593 |
| | SCL$_{ss}$ [17] | TMM22 | 0.562 | 0.538 | 0.550 | 0.599 | 0.577 | 0.588 | 0.613 | 0.579 | 0.596 |
| | OTPAL | - | **0.570** | **0.556** | **0.563** | **0.609** | **0.589** | **0.599** | **0.614** | **0.600** | **0.607** |

Table 3: Retrieval performance for mAP scores compared to existing methods on the XmediaNet dataset with data type-1.

| Type | Methods | Venue | 20% Label data | | | 40% Label data | | | 60% Label data | | |
|---|---|---|---|---|---|---|---|---|---|---|---|
| | | | I2T | T2I | Avg | I2T | T2I | Avg | I2T | T2I | Avg |
| US-CMR | DCCA [39] | arXiv15 | 0.206 | 0.226 | 0.216 | 0.239 | 0.226 | 0.232 | 0.243 | 0.229 | 0.236 |
| | DCCAE [35] | PMLR15 | 0.205 | 0.226 | 0.215 | 0.209 | 0.227 | 0.218 | 0.209 | 0.231 | 0.220 |
| S-CMR | DSCMR [47] | CVPR19 | 0.663 | 0.682 | 0.672 | 0.686 | 0.702 | 0.694 | 0.706 | 0.719 | 0.712 |
| | DAVAE [14] | MM20 | - | - | - | - | - | - | - | - | - |
| | PAN [43] | SIGIR21 | 0.451 | 0.485 | 0.468 | 0.456 | 0.486 | 0.471 | 0.457 | 0.491 | 0.474 |
| | C$^3$CMR [34] | MM22 | 0.542 | 0.548 | 0.545 | 0.576 | 0.586 | 0.581 | 0.595 | 0.601 | 0.598 |
| | ICMR-DCT [29] | TOMM23 | 0.573 | 0.544 | 0.558 | 0.610 | 0.567 | 0.588 | 0.586 | 0.595 | 0.590 |
| SS-CMR | SMLN [12] | AAAI20 | 0.609 | 0.654 | 0.631 | 0.619 | 0.638 | 0.628 | 0.608 | 0.638 | 0.623 |
| | SCL$_{ss}$ [17] | TMM22 | 0.636 | 0.648 | 0.642 | 0.672 | 0.682 | 0.677 | 0.675 | 0.685 | 0.680 |
| | OTPAL | - | **0.726** | **0.706** | **0.716** | **0.731** | **0.744** | **0.737** | **0.738** | **0.752** | **0.745** |

## 4.2 Implementation Details

We follow the image and text encoder networks utilized by CLIP [24] methods. Specifically, the visual encoder utilizes ResNet50 [9] as its base framework, integrating the style of the Transformer architecture [32]. Through a linear projection layer, it produces 1024-dimensional image representations. For the text transformer encoder, we first employ a lower-cased Byte Pair Encoding (BPE) with a vocabulary size of 49,152 words [27] to tokenize the textual descriptions. The textual descriptions are enclosed within [SOS] and [EOS] tokens indicating the beginning and end of the text sequence. The tokenized text is then fed into the transformer's module, and the textual features at the [EOS] position are normalized and processed by using a linear projection layer to output 1024-dimensional textual representations. To learn modality-shared features and obtain classification predictions for image and text modalities, we employ four fully connected layers with GELU activation function [10] for each modality, i.e., $1024 \rightarrow 2048 \rightarrow 1024 \rightarrow K$. The entire network is optimized by the Adam update rule [15] with the learning rate $10^{-3}$ and batch size 128. The training epoch is set to 400 for the XmediaNet dataset and 200 for the other datasets. In addition, we set the dropout ratio to 0.5, and the early stop to 20. For the hyper-parameters in OTPAL, we set $\alpha$, $\beta$, $\tau$ to $\{10, 5, 0.9\}$ on XmediaNet, $\{15, 1, 0.5\}$ on NUS-WIDE-10K, Wikipedia and Pascal Sentence. For other hyper-parameters, $T = 0.5$, and the number of nearest neighbors $k$ is set to 3 and 5 for Pascal-Sentence, and XMediaNet, respectively. The proposed model is trained on one 24GB Nvidia RTX A5000 GPU in PyTorch.

**Table 4: Average mAP scores for training data type-2 on two benchmark datasets.**

| Percentage | Pascal-Sentence | | | XmediaNet | | |
|---|---|---|---|---|---|---|
| | $SCL_{ss}$ | OTPAL1 | OTPAL | $SCL_{ss}$ | OTPAL1 | OTPAL |
| 10%L, 45%I, 45%T | 0.561 | 0.560 | 0.579 | 0.596 | 0.640 | 0.673 |
| 20%L, 40%I, 40%T | 0.633 | 0.635 | 0.650 | 0.642 | 0.692 | 0.710 |
| 30%L, 35%I, 35%T | 0.672 | 0.673 | 0.687 | 0.660 | 0.703 | 0.720 |
| 40%L, 30%I, 30%T | 0.678 | 0.679 | 0.692 | 0.677 | 0.716 | 0.729 |
| 60%L, 20%I, 20%T | 0.679 | 0.683 | 0.697 | 0.680 | 0.729 | 0.740 |

**Table 5: Ablation study: mAP scores of OTPAL and its three components on two datasets with 20 % labeled paired data.**

| Methods | NUS-WIDE-10K | | | XmediaNet | | |
|---|---|---|---|---|---|---|
| | I2T | T2I | Avg | I2T | T2I | Avg |
| OTPAL | 0.631 | 0.639 | 0.635 | 0.726 | 0.706 | 0.716 |
| OTPAL w/o $L_{MCIA}$ | 0.615 | 0.624 | 0.619 | 0.710 | 0.712 | 0.711 |
| OTPAL w/o $L_{pv}^l + L_{pt}^l$ | 0.586 | 0.603 | 0.594 | 0.680 | 0.682 | 0.681 |
| OTPAL w/o $L_{pv}^u + L_{pt}^u$ | 0.608 | 0.614 | 0.611 | 0.691 | 0.691 | 0.691 |

## 4.3 Evaluation Metric and Compared Methods

We evaluate the retrieval performance with the mean Average Precision (mAP), which is a widely-used performance evaluation criterion in cross-modal retrieval tasks [11, 13, 19]. In our experiments, we take images and texts as queries to calculate the cosine similarity to retrieve the relevant text and image samples, which can be denoted as Image2Text (I2T) and Text2Image (T2I). To evaluate the effectiveness of the OTPAL, we compare the proposed method with nine representative cross-modal retrieval baseline methods: unsupervised cross-modal methods (US-CMR): DCCA [39] and DCCAE [35], supervised methods (S-CMR): DSCMR [47], DAVAE [14] PAN [43], C³CMR [34], and ICMR-DCT [29], semi-supervised methods (SS-CMR): SMLN [12], SCL [17] and our method OTPAL. **Notice that all methods use the same feature encoder as our method.**

## 4.4 Comparison with State-of-the-Art Methods

Given that OTPAL is designed for partially aligned cross-modal retrieval, its performance is evaluated based on data type-1, where a certain proportion of labeled image-text pairs is retained, while the rest is randomly shuffled to construct unlabeled unpaired data. Tables 1-3 present the retrieval mAP scores under varying label ratios on three datasets, respectively. We use "- " for methods that do not provide source codes or results. As shown in these results, we can draw the following conclusions: 1) Training with partially aligned data may significantly harm the performance of cross-modal retrieval. As the proportion of labeled paired data decreases, the mAP scores of these methods will drop rapidly. 2) When labeled paired data is scarce, the retrieval performance of unsupervised methods does not surpass that of supervised methods. This is because, in partially aligned cross-modal retrieval, unsupervised methods are unable to utilize unlabeled aligned data to learn aligned representations between paired data. 3) Although the semi-supervised methods SMLN and SCL can achieve good performance, their improvement is limited. Mainly because they do not sufficiently exploit the discrimination and intrinsic correlations when encountering unlabeled unpaired data. Hence, their performance may be worse than some supervised methods. 4) Our method outperforms all

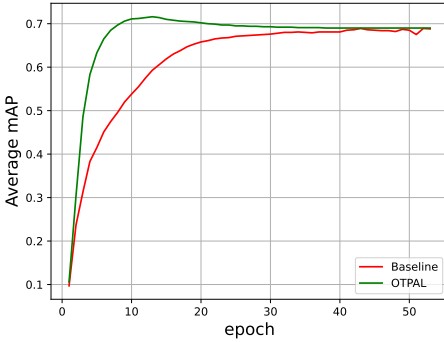

(a) XmediaNet

**Figure 4: Visual comparison of total OTPAL and Baseline on XmediaNet with 20% labeled paired data.**

existing SOTA methods on all datasets with varying proportions of labeled paired data, demonstrating the outstanding robustness of OTPAL in handling partially aligned data. Moreover, when the labeled paired data ratio is lower, the improvement of OTPAL is more evident, which can be observed in XmediaNet Table 3. In conclusion, the effectiveness of our approach can be attributed to the intrinsic connection established between labeled paired data and unlabeled unpaired data, which can further minimize the intra-class distance and improve model discrimination through optimal transport-based prototype alignment.

## 4.5 Ablation Studies

To validate the effectiveness of each module of our method, we conduct ablation studies for these modules, including the Dual Optimal Transport-based Prototype Alignment (DOTPA), and Prototype-based Neighborhood Feature Completion (PNFC). Experiments are conducted on Pascal-Sentence, NUS-WIDE-10K, and XmediaNet.

**Effectiveness of DOTPA.** Specifically, we ablate the contributions of three key components in our OTPAL, i.e., $L_{MCIA}$, $L_{pv}^l + L_{pt}^l$, and $L_{pv}^u + L_{pt}^u$. All compared methods are trained under 20% labeled paired data using the same settings as our OTPAL. The experimental results are reported in Table 5. From the results, the following observations can be drawn: 1) We can observe that the full OTPAL achieves the best performance, showing that all three components are important to improve the performance towards partially aligned data. Furthermore, in the visualization comparison between total OTPAL and Baseline ($L_{MCIA}$), we can observe that the former converges to the best performance faster than the baseline in Fig. 4. 2) Both $L_{pv}^l + L_{pt}^l$ and $L_{pv}^u + L_{pt}^u$ help to improve the performance. They are complementary to each other. $L_{pv}^l + L_{pt}^l$ can make the features more discriminative so that samples of different categories are easier to distinguish on labeled paired data, while $L_{pv}^u + L_{pt}^u$ can significantly minimize intra-class distances to improve the discriminability and modal invariance on unlabeled unpaired data.

**Effectiveness of PNFC.** To evaluate the robustness of our OTPAL towards incomplete partially aligned data, we compare it with the SOTA method SCL under data type-2. The experimental results are reported in Table 4. OTPAL1 represents the OTPAL without the Prototype-based Neighborhood Feature Completion module

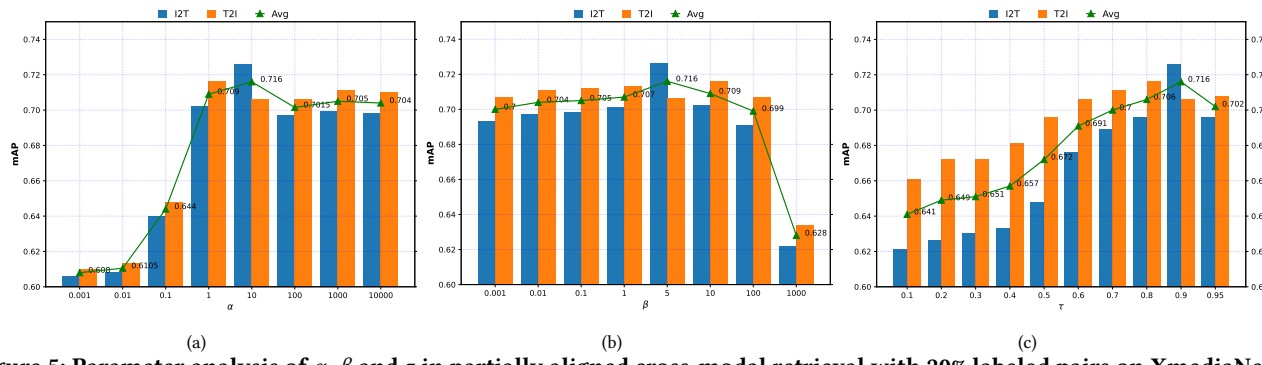

(a)                                        (b)                                        (c)

Figure 5: Parameter analysis of $\alpha$, $\beta$ and $\tau$ in partially aligned cross-modal retrieval with 20% labeled pairs on XmediaNet.

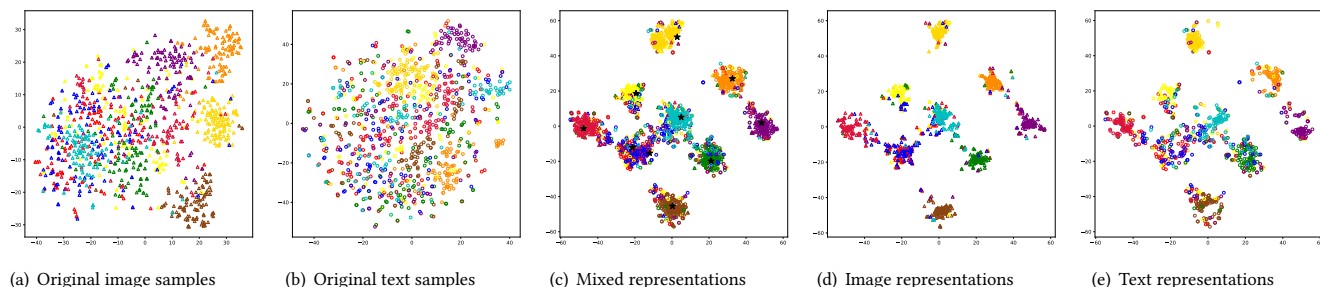

(a) Original image samples     (b) Original text samples     (c) Mixed representations     (d) Image representations     (e) Text representations

Figure 6: Visualization for testing data on NUS-WIDE-10K by t-SNE [31]. Triangles denote image features, circles denote text features, and stars denote shared semantic prototypes. Features of the same class are indicated with the same color.

(PNFC). By comparing OTPAL and OTPAL1, it is shown that exploring incomplete modality data is very important for incomplete part-aligned cross-modal retrieval. By comparing OTPAL with other methods, it shows that our method leverages cross-modal neighbors guiled by shared prototype information to generate completion features, alleviate the impact of performance degradation caused by missing data, and further improve the robustness of our OTPAL.

### 4.6 Parametric Sensitivity Analysis

The main hyper-parameters involved are $\alpha$, $\beta$ and $\tau$. The objective function contains two parameters $\alpha$ and $\beta$, which control the contributions of different loss functions. We evaluate their influences on the XmediaNet dataset, and report the results in Fig. 5 $(a) - (c)$. These experiments are conducted under the 20% labeled paired multi-modal data scenario. Specifically, to accurately reflect the performance variations affected by hyper-parameters, we verify the effect of one hyper-parameter while keeping other hyper-parameters unchanged. For $\alpha$, we set the range of the hyper-parameter in Eq. (22) as {0.001, 0.01, 0.1, 1, 10, 100, 1000, 10000}, and we can observe that the mAP scores in both directions tend to be stable when $1 < \alpha < 10000$. This also proves that OTPAL can keep high performance over a wide range of $\alpha$. For $\beta$ in Eq. (7), the changing trend of mAP score is opposite to $\alpha$. The performance is stable in a larger range of {0.001 100}, and it drops sharply when it exceeds 100. For $\tau$ below Eq. (14), the mAP first increases with the growth of $\tau$, and then begins to decrease after $\tau$ surpasses a threshold 0.9, which shows that a suitable threshold can effectively select reliable prototypes that perform accurate alignment between instances and prototypes. In conclusion, OTPAL is relatively insensitive to the selection of $\alpha$ and $\beta$, but sensitive to the choice of $\tau$.

### 4.7 Feature Visualization

We visualize the learned visual and textual representations produced by the modality-specific layer using t-SNE in Fig. 6. The original image and text features are from the testing set of NUS-WIDE-10K. We contrast t-SNE 1024-dimensional embeddings of the original features (Fig. 6 $(a) - (b)$) and the learned visual and textual features (Fig. 6 $(c) - (e)$). We can clearly see that the original features of different modalities present different spatial distributions, and instances from different categories cannot be separated (Fig. 6 $(a) - (b)$). Besides, we can observe that both the image and text features of OTPAL exhibit consistent semantic distribution and are aligned closely (Fig. 6 $(c) - (e)$), which validates the effectiveness of our method in learning modality invariance and semantic discrimination. This can be attributed to the fact that shared prototypes can establish intrinsic correlations between images and texts and better characterize global semantic information (Fig. 6 $(c)$ stars).

### 5 CONCLUSION

To tackle the challenges of partially aligned cross-modal retrieval in real-world scenarios, we propose a novel optimal transport-based prototype alignment learning for robust cross-modal retrieval. In particular, to learn modality discriminative and invariant representations while minimizing intra-class distance, we propose a dual optimal transport-based prototype alignment strategy. Further, to deal with the problem of incomplete partially aligned data, a prototype-based neighbor feature completion is proposed, which can extend our method to more realistic scenarios. Extensive experiment results demonstrate the superiority of our model in partially aligned cross-modal retrieval by comparing it with SOTA methods.

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
