# OpenReview forum: "Partially Aligned Cross-modal Retrieval via  Optimal Transport-based Prototype Alignment Learning"
_acmmm.org/ACMMM/2024/Conference — MM2024 Poster_

### Official Review · Reviewer_LdL5 · 2024-05-16

**Rating:** 4
**Confidence:** 3

**Summary:**

This paper investigates the problem of cross-modal retrieval with unlabeled and unpaired data, and proposes a reasonable approach OTPAL for this problem.

**Strengths:**

1. The motivation is clear and the method is reasonable.
2. The experiments are comprehensive and the results demonstrate improved performance.

**Limitations:**

1. Lack of latest and sota baseline methods such as RONO, CLF, ALGCN, GNN4CMR, etc.
2. The proposed method is reasonable but not novel. It seems that the overall framework is just a combination of some well-known techniques.

**Suitability:**

3

---

### Official Review · Reviewer_k2sT · 2024-05-23

**Rating:** 4
**Confidence:** 4

**Summary:**

● This paper introduces an Optimal Transport-based Prototype Alignment Learning (OTPAL) to encounter the problem of partially alignment multi-modal pairs in real-world scenarios.
● The paper categorizes the partially aligned multi-modal pairs into unlabeled unpaired, and unlabeled incompleted data.
● The authors propose a dual optimal transport-based prototype alignment (DOTPA) to resolve the optimal transport problem between data (image or text) and prototypes, enabling cross-modal alignment.
● For the unlabled incomplete data, this paper introduces a prototype-based neighborhood feature completion (PNFC) method to generate the representation of the missing modality. The prototypes are adopted to provide similar semantics to recover the missing modality.

**Strengths:**

● This paper explicitly illustrates the situation of the partially aligned multi-modal pairs in the real world and proposes corresponding approaches to resolve the challenges in each type of partially aligned data.
● The methods are clear and easy to follow.
● The experiments are sufficient and the proposed methods maintain a considerable performance.

**Limitations:**

● In section 2, it lacks of the review of the incomplete multi-modal learning scenarios.
● The compared baselines are out of date. The authors should compare with the recent research on partially aligned cross-modal retrieval.
● The author should provide the experiment results on 100% labeled data to illustrate the necessity of this paper.
● The font size in Fig.5 is too small to read.

**Suitability:**

3

---

### Official Review · Reviewer_EE7t · 2024-05-24

**Rating:** 3
**Confidence:** 4

**Summary:**

The paper introduces a novel approach for partially aligned cross-modal retrieval called Optimal Transport-based Prototype Alignment Learning (OTPAL). This method tackles the challenge of unannotated, unaligned cross-modal samples by establishing matching correlations between shared prototypes and samples using the optimal transport algorithm. Additionally, OTPAL introduces a scalable prototype-based neighbor feature completion method to handle incomplete multimodal data, thereby enhancing the robustness of cross-modal retrieval systems in practical applications. While the effectiveness of OTPAL is demonstrated, the paper lacks sufficient discussion on related works in the area of partially unaligned or incomplete multi-modal learning.

**Strengths:**

1. The paper presents a novel setting for the cross-modal retrieval task.
2. Extensive experimental results verify the effectiveness of the proposed method.

**Limitations:**

1. My major concern is the missing connections with the existing works on partially unaligned and incomplete multi-modal learning studies which are highly related to this work. For example, [A] for partially view-aligned learning, [B for incomplete multi-view learning, [C] for the unified solution of partially aligned and incomplete multi-view learning. [A] Partially View-aligned Representation Learning with Noise-robust Contrastive Loss [B] Incomplete Multi-view Clustering via Prototype-based Imputation, IJCAI2023 [C] Robust Multi-view Clustering with Incomplete Information.
2. The optimal transport algorithm may involve high computational complexity. It is recommended to analyze the time complexity or present the time cost for clarity.
I will consider raising my rating if the authors' feedback addresses my concerns.

**Suitability:**

3

---

### Official Review · Reviewer_E2nH · 2024-05-25

**Rating:** 3
**Confidence:** 4

**Summary:**

This study develops a method called Optimal Transport-based Prototype Alignment Learning (OTPAL). It classifies these multi-modal pairs into two categories: unlabeled unpaired data and unlabeled incomplete data. To solve the optimal transport problem across different modalities (such as images and text) and their prototypes, the authors put forward a method named Dual Optimal Transport-based Prototype Alignment (DOTPA), which promotes alignment across these modalities. Addressing the issue of unlabeled incomplete data, the study proposes a Prototype-based Neighborhood Feature Completion (PNFC) approach. This method utilizes prototypes to approximate and restore the semantics of the absent modalities.

**Strengths:**

This document thoroughly describes the condition of partially aligned multi-modal pairs in real-world settings and suggests specific strategies to overcome the obstacles presented by each type of partially aligned data.

**Limitations:**

- The authors may violate the submission policy in the block of the bottom left corner of page 1.
- The authors do not provide experimental comparison of 100% labeled data, which makes it difficult to conclude that the improvement of experimental results is from the contribution of this paper, rather than the backbone.
- The baselines are out of date. The authors should provide the baselines more recently.

**Suitability:**

3

---

### Meta-Review · Area_Chair_J5Ut · 2024-06-29

**Recommendation:** Accept (Poster)
**Confidence:** 5

**Metareview:**

The paper introduces a novel approach, called Optimal Transport-based Prototype Alignment Learning (OTPAL), for partially aligned cross-modal retrieval. The initial ratings were two Borderline Reject and two Borderline Accept. After the rebuttal, two reviewers changed their recommendation ratings, and all reviewers reached a consensus to Borderline Accept. The authors are suggested to include a more comprehensive review on incomplete multi-modal/view learning and unaligned multi-modal/view learning in their final version.